# Homocysteine Metabolism Pathway Is Involved in the Control of Glucose Homeostasis: A Cystathionine Beta Synthase Deficiency Study in Mouse

**DOI:** 10.3390/cells11111737

**Published:** 2022-05-25

**Authors:** Céline Cruciani-Guglielmacci, Kelly Meneyrol, Jessica Denom, Nadim Kassis, Latif Rachdi, Fatna Makaci, Stéphanie Migrenne-Li, Fabrice Daubigney, Eleni Georgiadou, Raphaël G. Denis, Ana Rodriguez Sanchez-Archidona, Jean-Louis Paul, Bernard Thorens, Guy A. Rutter, Christophe Magnan, Hervé Le Stunff, Nathalie Janel

**Affiliations:** 1Unité de Biologie Fonctionnelle et Adaptative, Université Paris Cité, CNRS, 75013 Paris, France; mk.meneyrol@gmail.com (K.M.); jessica.denom@u-paris.fr (J.D.); nadim.kassis@u-paris.fr (N.K.); stephanie.migrenne@u-paris.fr (S.M.-L.); fabrice.daubigney@univ-paris-diderot.fr (F.D.); raphael.denis@u-paris.fr (R.G.D.); christophe.magnan@u-paris.fr (C.M.); nathalie.janel@u-paris.fr (N.J.); 2Institut Cochin, Université Paris Cité, INSERM U1016, CNRS UMR 8104, 75014 Paris, France; latif.rachdi@inserm.fr (L.R.); fatna.makaci@inserm.fr (F.M.); 3Section of Cell Biology and Functional Genomics, Division of Diabetes, Endocrinology and Metabolism, Department of Metabolism, Digestion and Reproduction, Imperial College London, London W12 0NN, UK; e.georgiadou@imperial.ac.uk (E.G.); g.rutter@imperial.ac.uk (G.A.R.); 4Center for Integrative Genomics, University of Lausanne, 1015 Lausanne, Switzerland; ana.rodriguez-sanchez-archidona@chuv.ch (A.R.S.-A.); bernard.thorens@unil.ch (B.T.); 5Georges Pompidou European Hospital Department of Biochemistry, Assistance Publique-Hôpitaux de Paris (AP-HP), 75013 Paris, France; jean-louis.paul@egp.aphp.fr; 6Cardiometabolic Axis, CR-CHUM, Université de Montréal, Montréal, QC H3T 1J4, Canada; 7Lee King Chian Medical School, Nanyang Technological University, Singapore 639798, Singapore; 8Institut des Neurosciences Paris-Saclay, CNRS UMR 9197, Université Paris-Saclay, 91400 Saclay, France; hlestunff62@gmail.com

**Keywords:** autonomic nervous system, insulin secretion, hyperhomocysteinemia, type 2 diabetes

## Abstract

Cystathionine beta synthase (CBS) catalyzes the first step of the transsulfuration pathway from homocysteine to cystathionine, and its deficiency leads to hyperhomocysteinemia (HHcy) in humans and rodents. To date, scarce information is available about the HHcy effect on insulin secretion, and the link between CBS activity and the setting of type 2 diabetes is still unknown. We aimed to decipher the consequences of an inborn defect in CBS on glucose homeostasis in mice. We used a mouse model heterozygous for CBS (CBS+/−) that presented a mild HHcy. Other groups were supplemented with methionine in drinking water to increase the mild to intermediate HHcy, and were submitted to a high-fat diet (HFD). We measured the food intake, body weight gain, body composition, glucose homeostasis, plasma homocysteine level, and CBS activity. We evidenced a defect in the stimulated insulin secretion in CBS+/− mice with mild and intermediate HHcy, while mice with intermediate HHcy under HFD presented an improvement in insulin sensitivity that compensated for the decreased insulin secretion and permitted them to maintain a glucose tolerance similar to the CBS+/+ mice. Islets isolated from CBS+/− mice maintained their ability to respond to the elevated glucose levels, and we showed that a lower parasympathetic tone could, at least in part, be responsible for the insulin secretion defect. Our results emphasize the important role of Hcy metabolic enzymes in insulin secretion and overall glucose homeostasis.

## 1. Introduction

The cystathionine beta synthase enzyme (CBS) converts serine and homocysteine into cystathionine as the first step of the transsulfuration pathway, and is part of the metabolic pathway that converts methionine into cysteine [1,2]. A deficiency in CBS activity leads to markedly increased plasma homocysteine (Hcy) levels, namely severe hyperhomocysteinemia (HHcy), in human and rodents. Plasma physiological levels of Hcy in humans are commonly considered normal between 5 and 15 μmol/L while HHcy is considered as moderate when ranging from 15 to 30 μmol/L, as intermediate for values between 30 and 100 μmol/L, and severe for values exceeding 100 μmol/L.

Although plasma Hcy levels can significantly vary among the different populations according to their dietary habits, HHcy is considered as a biomarker for several pathologies. HHcy is associated with cardiovascular risk such as thrombosis and cardiac hypertrophy, neuropsychiatric illness, and hepatic steatosis and fibrosis (leading to liver failure) [3]. The loss of CBS activity due to mutation in humans leads to a disease named classical homocystinuria (HCU), the most frequent inborn error of sulfur amino acid metabolism, with consequent accumulation of Hcy in the plasma and tissues. Patients have skeletal and connective tissue defects, thrombosis and developmental delay, intellectual disability and seizures, and fatty liver [4]. To ameliorate these pathologies, pharmacological enzyme replacement and gene transfer therapies are being developed. In mice, Cbs KO is neonatally lethal, whereas the administration of an enzyme therapy [4] has been shown to rescue KO mice by preventing liver disease when given during five weeks of life.

CBS is mainly expressed in the liver, which is the key organ maintaining methionine/Hcy homeostasis; HHcy has frequently been associated with non-alcoholic fatty liver disease (NAFLD) [5] and type 2 diabetes (T2DM). Recent data show that folic acid supplementation may reduce Hcy levels in patients with T2DM and produce a better glycemic control compared to the placebo [6]. However, it is still unclear whether predisposition to HHcy could contribute to the severity of T2DM. The relationship between HHcy and obesity has not been totally elucidated. Moreover, the existence of a causal link between moderate to intermediate HHcy due to a decreased CBS activity and alteration of glucose homeostasis is unclear. Several studies from Wang’s laboratory demonstrated that HHcy induced insulin resistance in adipose tissue by promoting ER stress and upregulating resistin production, however, these studies used a model of short-term Hcy administration in drinking water in WT mice, which only reached moderate HHcy [7].

In this study, we used mice heterozygous for the targeted disruption of the Cbs gene (CBS+/−) and wild type (CBS+/+) mice on the same background. As shown by Watanabe and colleagues, CBS+/− mice have a 50% reduction in CBS mRNA and enzyme activity in the liver and twice the normal plasma Hcy levels [8]. We supplemented the mice with 0.5% L-methionine in drinking water to induce intermediate HHcy in CBS+/− mice, around 6-fold above the normal [9,10], and a specific cohort was fed with a high-fat diabetogenic diet (HFD) [11] in order to study the combined effects of intermediate HHcy and disturbed glucose homeostasis.

We demonstrate that the constitutive deficiency of CBS in CBS+/− mice, with methionine supplementation and under a high-fat diet (HFD), led to a defect in insulin secretion compensated with increased insulin sensitivity. The evidence shows that a lower parasympathetic tone could, at least in part, be responsible for this phenotype.

## 2. Materials and Methods

### 2.1. Mouse Phenotyping

All procedures were carried out in accordance with the ethical standards of the French and European regulations (European Communities Council Directive, 86/609/EEC). Animal use and procedures were approved by the Ethics committee of the University of Paris and by the French Ministry of Research (approval # 8728-20 16082615248655 v3). Mice were housed in a controlled environment with unlimited access to food and water on a 12-h light/dark cycle. The number of mice and suffering were minimized as much as possible. Mice heterozygous for targeted disruption of the Cbs gene (CBS+/−), on a C57BL/6 background, were obtained by mating male CBS+/− mice with female wild-type C57BL/6 (CBS+/+) mice. We used male mice heterozygous for CBS (CBS+/−) that presented a 2-fold increase in the Hcy levels [10]. Another group of male CBS+/− mice and controls were supplemented with 0.5% methionine in drinking water to induce intermediate HHcy; the latter were also submitted to a high-fat diet (SAFE 230 HF, Augy, France). We characterized the mice for their food intake, body weight gain, body composition using EchoMRI (Houston, TX, USA), glucose homeostasis, CBS activity in the liver, and Hcy levels in plasma. We checked that the methionine in drinking water did not provoke any aversion. In a specific cohort, calorimetric analyses for the energy expenditure, O_2_ consumption and CO_2_ production, respiratory exchange rate, food intake, and spontaneous locomotor activity (beam breaks per hour) were monitored automatically using calorimetric cages as described [12]. The estimation of the basal metabolism was calculated according to Peterfì et al. [13].

### 2.2. CBS Enzyme Activity Assays

Determination of the CBS activity was assayed on 400 µg of total proteins obtained from liver samples, as described previously [14]. Proteins were incubated for 1 h at 37 °C with 1 mm propargylglycine, 0.2 mm pyridoxal phosphate, 10 mM L-serine, 10 mM DL-Hcy, and 0.8 mm S-(5′-adenosyl)-L-methionine using the DTNB (5,5′-dithiobis-(2-nitrobenzoic acid)) based-assay. The reaction was monitored at 37 °C by measuring the absorbance at 412 nm every 10 min using a spectrophotometer (Lambda XLS, PerkinElmer, MA, USA). All of the chemical products were obtained from Sigma (Sigma-Aldrich, Saint-Quentin-Fallavier, France).

### 2.3. Plasma Hcy Assays

Blood samples were collected into tubes containing a 1/10 volume of 3.8% sodium citrate, and immediately placed on ice. Plasma was isolated by centrifugation at 2500× *g* for 15 min at 4 °C. Plasma total Hcy, defined as the total concentration of Hcy after quantitative reductive cleavage of all disulfide bonds, was assayed using the fluorimetric high-performance liquid chromatography (HPLC) method, as previously described [15].

### 2.4. Glucose and Insulin Tolerance Tests

For the glucose tolerance tests, mice were food deprived for 5 h prior to an oral administration of 2 g/kg of 30% glucose. Blood was sampled from the tail vein at 0, 15, 30, 60, 90, and 120 min in order to assay the glucose concentration with a glucometer (A. Menarini Diagnostics, France). At 0, 15, and 90 min, blood was taken to measure the plasma insulin with the Ultra-Sensitive Mouse Insulin ELISA Kit #90080 (Crystal Chem Inc., Elk Grove Village, IL, USA).

For the insulin tolerance tests, mice were food deprived for 5 h prior to the intra-peritoneal administration of insulin (Novorapid^®^) at 0.5 UI/Kg. Blood was sampled from the tail vein at 0, 15, 30, 45, 60, 90, and 120 min in order to assay the glucose concentration with a glucometer (A. Menarini Diagnostics, Rungis, France).

### 2.5. Immunohistochemistry and Quantification

Paraffin sections were processed for immunohistochemistry as previously described [16]. Slides were scanned using a Panoramic 250 Flash II scanner (3DHISTECH). Images were analyzed for β-cell area and the pancreatic area using inForm Tissue Finder software v2.3.0, Akoya Biosciences, MA, USA. Quantification of insulin staining was performed on five equally separated pancreas sections. The percentage of the β-cell area in each pancreatic section was determined by dividing the total area of insulin-positive cells by the surface area of the section and the β-cell mass was calculated by multiplying the pancreas weight by the % area of the β-cells.

### 2.6. Islet Isolation and Culture

Each pancreas was distended by the intraductal injection of a collagenase solution, at 7500 U/pancreas (Collagenase type P, Roche, Bâle, Switzerland). Collagenase was dissolved in 5 mL of Hanks’ balanced salt solution (HBSS Gibco, Invitrogen, Carlsbad, CA, USA) with HEPES 10% (Gibco, Invitrogen). The pancreases were incubated in a 50 mL conical tube at 37 °C for 15 min. Incubation tubes were then shaken slowly for 30 s and 25 mL of cold buffer containing HBSS, BSA, and HEPES was added to stop the digestion (HBSS: Gibco, Invitrogen; BSA, Interchim SA, Montluçon, France). The digested pancreases were washed with buffer three times by centrifugation (1200 rpm, 2 min, 4 °C). The islets were purified on a four-layer density gradient of Histopaque 1119 (Sigma-Aldrich) and HBSS, in a 15 mL conical tube. After centrifugation (2000 rpm, 25 min, 4 °C), all islets from the various layers were collected and placed in a 50 mL conical tube before centrifugation (2500 rpm, 10 min, 4 °C). Pellets containing the islets were resuspended in ice-cold buffer before they were placed in Petri dishes.

### 2.7. Static Batch Incubation

After isolation, the floating islets were incubated overnight at 37 °C in a humidified atmosphere and 5% CO_2_ in a RPMI media containing (RPMI medium 1640: Gibco, Invitrogen, France; 10% inactivated fetal calf serum (FCS), 1% HEPES, 11 mmol/L glucose (D-Glucose 45%, Sigma Aldrich, France), 100 IU/mL penicillin, and 100 mg/mL streptomycin).

Next, batches of five size-matched islets were pre-incubated in KRBH-0.1% BSA with 2.8 mm of glucose for 30 min, followed by 60 min incubation at 37 °C in KRBH-0.1% BSA buffer containing 2.8 mm or 16.7 mm of glucose to measure the glucose-induced insulin secretion. The supernatant (500 µL) was immediately collected and stored at −20 °C until being assayed for insulin by ELISA (ultra-sensitive mouse insulin ELISA Kit; Crystal Chem, #90080, the Netherland). The islets were homogenized in protein extraction buffer and stored until insulin content determination by ELISA (ultra-sensitive mouse insulin ELISA Kit; Crystal Chem, #90080, Netherland), as previously described [17].

### 2.8. Measurement of Intracellular Free Ca^2+^ and Pearson (R)-Based Connectivity and Correlation Analyses

Functional multicellular Ca^2+^-imaging was performed on isolated islets as previously described [17]. Correlation analyses were performed between beta-cell pairs and their extracted fluorescent Ca^2+^ traces over time in each imaged islet in MATLAB using a custom-made script, as previously described [18].

### 2.9. RNA Extraction, cDNA Synthesis, and Real-Time PCR Using SYBR-Green Chemistry

Total RNA was isolated from pancreatic islets using the RNeasy Lipid Kit (Qiagen, Hilden, Sweden). The concentration of the RNA samples was ascertained by measuring the optical density at 260 nm. The quality of RNA was verified by the optical density absorption ratio OD 260 nm/OD 280 nm. To remove the residual DNA contamination, the RNA samples were treated with RNAse-free DNAse (Qiagen) and purified with an RNeasy mini column (Qiagen). For each sample, 4 µg of total RNA from each sample was reverse transcribed with 200 U of M-MLV Reverse Transcriptase (Invitrogen, Life Technologies, Carlsbad, CA, USA) using random hexamer primers. Real-time quantitative PCR amplification reactions were carried out in a LightCycler 480 detection system (Roche) using the LightCycler FastStart DNA Master plus SYBR Green I Kit (Roche). Primer sequences are given in Appendix A. For each reaction, 40 ng of reverse transcribed RNA was used as the template. All reactions were carried out in duplicate with a no template control. The PCR conditions were: 95 °C for 5 min, followed by 45 cycles of 95 °C for 10 s, 60 °C for 10 s, and 72 °C for 10 s. The mRNA transcript level was normalized against rpL19. Primer sequences were 5′GGGCAGGCATATGGGCATA3′ (sense primer) and 5′GGCGGTCAATCTTCTTGGATT3′ (antisense primer) for rpL19. To compare the target gene level, relative quantification was performed as outlined in Pfaffl et al. [19].

### 2.10. Correlation Studies in Liver and Islets

Correlations between the expression of CBS in the mouse islets and liver, and metabolic phenotypes were carried out on the basis of the database developed within the framework of the European program IMI Rhapsody and according to a previously described methodology [11]. Briefly, male mice from three commonly used non-diabetic mouse strains C57BL6/J, DBA2/J, and BalbC were fed a high-fat or regular chow diet for one month. Tissues were collected and pancreatic islets were extracted and phenotypic measurements (basal glycemia and insulinemia, oral GTT, ITT) were performed at 2 days, 10 days, and 30 days to assess diabetes progression. RNA-Seq was performed on the islet tissue at each time-point and integrated with the phenotypic data in a network-based analysis [11].

### 2.11. Network Construction

The network was created with six genes involved in the homocysteine metabolism expressed in the pancreatic islets, their gene co-expression modules, the selected annotated KEGG pathways/GO functional categories of the modules, and the phenotypic traits represented as nodes. The homocysteine pathway genes were Cbs, dual specificity tyrosine phosphorylation regulated kinase 1A (Dyk1a), methionine synthase (Mtr), methionine adenosyltransferase 1A (Mat1a), methylenetetrahydrofolate reductase (Mthfr), and S-adenosyl-L-homocysteine hydrolase (Ahcy). These genes belonged to the modules ‘blue’, ‘turquoise’, ‘turquoise’, ‘grey’, ‘brown’, and ‘yellow’, respectively. The gene *Mat1a* was not assigned to any module (‘grey’ module). Annotated pathways were selected as those related to metabolism or with relevance in the context of T2D.

The edges of the network (interactions) represent the Spearman correlations between the nodes and were assigned between the network nodes as follows: A general edge if the correlation between the nodes had an absolute Spearman correlation coefficient of 0.65 or bigger (|rho|≥0.65) and a *p*-value ≤ 0.05; and a specific edge between the six genes and the phenotypic traits, if the correlation between the nodes had a |rho| ≥ 0.40 and a *p*-value ≤ 0.05.

The gene *Mthfr* was not included in the final network since its correlations with the traits were |rho| ≤ 0.40: consequently, the module ‘brown’ was also not included in the final network. The Spearman correlations were computed using the R software and the final network was exported to the Gephi 0.9.2 software for visualization purposes.

The count matrix was normalized using the trimmed mean (TMM) normalization method implemented within the edgeR R package [20].

## 3. Results

### 3.1. Determination of Moderate to Intermediate HHcy Effects on the Body Weight and Glucose Homeostasis In Vivo

To determine the consequences of a moderate HHcy on glucose homeostasis, we used CBS+/− mice fed with a regular chow diet. As displayed in Figure 1A, CBS is the enzyme catalyzing the conversion of homocysteine into cystathionine, thus a CBS deficiency leads to increased homocysteine levels to reach a moderate HHcy, as has already been demonstrated (cf. Figure 1B) [21]. In order to increase homocysteinemia and reach an intermediate HHcy, mice were supplemented with methionine in drinking water [21] and other cohorts were also subjected to a high-fat diet. Figure 1C shows that liver CBS activity was significantly lower in the CBS+/− mice regardless of the treatment or diet, and Table 1 displays the Hcy levels in the different groups.

#### 3.1.1. Genotype-Only Effect (Moderate HHcy) on Glucose Homeostasis

Body weight and body composition were similar in the CBS+/− vs. CBS+/+ mice (Table 1). The CBS+/− mice showed normal glucose tolerance and insulin sensitivity (Figure 2A−D) but have a lower insulin secretion in both the basal condition (*p* = 0.06, and 15 min after the glucose administration (*p* = 0.001), as shown in Figure 2E. In vitro insulin secretion showed similar levels at both 2.8 mm and 16.7 mm glucose (data not shown).

#### 3.1.2. Genotype and Methionine Effect (Intermediate HHcy) on Glucose Homeostasis

Under CD, body weight was slightly lower (NS) in CBS+/− mice supplemented with methionine (Met) for 5 mo compared to the CBS+/+ Met-supplemented mice, while the body composition was unchanged (Table 1). CBS+/− Met mice showed lower insulin secretion at a glucose stimulated state while the insulin sensitivity and glucose tolerance remained unchanged (Figure 3, Left panel, A, C, E, G, I). Under the HFD, the body weight was slightly lower in the CBS+/− mice due to the lower fat mass (NS). CBS+/− mice under methionine and HFD showed lower insulin secretion (both basal and glucose stimulated) while insulin sensitivity is significantly improved compared to respective controls (Figure 3 Right panel, B, D, F, H, J).

We reproduced the same finding on the lower insulin secretion in CBS+/− mice with methionine in drinking water for 3 mo and fed with a HF diet for 2 mo (CBS+/− Met-HF), after both the oral glucose tolerance test or ip. glucose tolerance test, suggesting that the incretin pathway was not involved in the phenotype (Appendix A). Hyperinsulinemic euglycemic clamps performed in CBS+/− Met/HF mice and the controls showed a tendency to overall improved insulin sensitivity as measured by the glucose infusion rate, and a significant increase in glycolytic muscle (elongator digitorum longus, EDL) glucose uptake following radio-labelled 2-deoxy-glucose administration (Appendix A).

### 3.2. Determination of Islets Insulin Secretion, Calcium Imaging, and Cell Connectivity

Although the gene expression study in the pancreatic islet showed a lower expression for transcription factors involved in beta-cell differentiation, in particular, Pdx1, which was 50% decreased in CBS+/− (Table 2), the pancreas histological analysis showed a similar beta-cell mass and unchanged insulin/total pancreas ratio in the CBS+/− mice compared to the WT, after Met-HF treatment (Figure 4A,B). Of note, the alpha cell mass was similar in both groups (data not shown). We then investigated in the vitro insulin secretion. As shown in Figure 4C, in vitro insulin secretion was unchanged in the CBS+/− Met-HF mice compared to their controls. However, after normalization with the insulin content of the islets, insulin secretion was lower at low glucose (2.8 mm), and similar to the controls at 16.7 mm, showing that the islets were able to respond to glucose (Figure 4D). Cytosolic Ca^2+^ imaging revealed that islets from the CBS+/− mice had an enhanced oscillatory response in intracellular free Ca^2+^ concentration during stimulation with 17 mm glucose in comparison to the CBS+/+ mice (Figure 4E−F). In contrast, the intracellular Ca^2+^ increase in response to depolarization induced by a short pulse of KCl was similar in both groups (Figure 4G). The latter finding supports the idea that increased Ca^2+^ signals in the CBS+/− mice may be mediated by changes in the glucose metabolism, upstream of the Ca^2+^ influx [22]. Accordingly, the mRNA encoding glucose transporter GLUT2 was found to be increased in the islet of the CBS+/− mice (Table 2), which could be related to increased glucose metabolism, as several potential mechanisms mediating the response of islet GLUT2 to glucose have been investigated and shown that glucose metabolism is instrumental in the stimulatory effects of glucose on beta-cell GLUT2 mRNA accumulation [23]. The proportion of interconnected beta-cells [24,25] increased uniformly in response to glucose increases (from 3 to 17 mm) or during KCl stimulation in both groups, as shown in Figure 4H.

These findings indicate that islets from the CBS+/− mice had a higher intrinsic ability to respond to glucose which, interestingly, did not affect the beta-cell-to-beta-cell coupling when compared to the CBS+/+ mice. Therefore, decreased insulin secretion in response to glucose observed in vivo is unlikely to be linked to a functional deficiency in glucose responses, but to responses to stimulation by other agents.

### 3.3. Differential Modulation of the Autonomic Nervous System in CBS+/− Mice

In vivo nervous recordings in anaesthetized mice showed a defect of parasympathetic activation in response to a glucose load, while basal activity remained unchanged (Figure 5A). This defect is consistent with a decreased insulin secretion in response to glucose as the parasympathetic nervous system (PNS) is known to stimulate anabolism and insulin secretion, while sympathetic nervous system (SNS), in contrast, stimulates catabolism and inhibits insulin secretion. Yohimbine (YOH) is a sympatholytic compound that decreases the ratio between the SNS and PNS, and its administration before OGTT led to an improved glucose tolerance in both the CBS+/+ and CBS+/− mice, however, the effect was stronger in the CBS+/+ mice where the slope of the blood glucose over time was significantly decreased over the 0–30 min period (Figure 5 B–D). In addition, the 2-way ANOVA of the OGTT for the comparison between the CBS+/+ Saline and CBS+/+ YOH returned a *p* < 0.0001, whereas the 2-way ANOVA for CBS+/− Saline and CBS+/− YOH was *p* = 0.0656, N.S. The latter finding is consistent with the low PNS tone. Carbachol, a cholinergic agonist that stimulates the parasympathetic system, also differentially modulates the glucose kinetics when given before OGTT (CARB in Figure 5). In the CBS+/+ mice, glucose tolerance was significantly improved following carbachol administration while it had no significant effect on the CBS+/− mice, suggesting an impairment in the parasympathetic activation (Figure 5 E–G), which was consistent with the decreased expression in the M3 muscarinic receptors in the CBS+/− islets (Table 2). Indeed, the 2-way ANOVA of the OGTT for the comparison between CBS+/+ Saline and CBS+/+ CARB returned a *p* = 0.0028, whereas the 2-way ANOVA for CBS+/− Saline and CBS+/− CARB was *p* = 0.2172, N.S.

Of note, intensive calorimetric analysis on both the Met-HF CBS+/− and CBS+/+ mice showed no significant differences with regard to the change in the energy expenditure, estimated resting metabolism, RER, nor locomotor activity at any time point. Food intake was monitored throughout the experiment, but due to the high spillage of the high-fat regimen, the data were not suitable to perform the analysis (Appendix A).

### 3.4. Islet Expression of CBS Pathway Genes Correlates with Glucose Homeostasis Traits in a Multi-Strain Study

Metabolic phenotyping measurements in high-fat-fed male mice from three commonly used non-diabetic mouse strains, C57BL/6J, BALB/cJ, and DBA/2J, which were recorded at 2 days, 10 days, and 30 days to evaluate pre-diabetes progression [26]. To identify the metabolic phenotype-related genes, RNA sequencing was performed on the pancreatic islets from these mice, which express CBS and several genes of the homocysteine metabolic pathway. Transcriptomic analysis was performed at each time-point and integrated with the phenotypic data. Our results identified the CBS gene as being one of several genes related to insulin resistance and basal insulinemia (Figure 6A,B). Its expression in islets was significantly and positively correlated with both insulin secretion in the basal condition (5 h food deprivation) and insulin resistance (Spearman correlations *p*-values ≤ 0.05). In the liver, the CBS expression positively correlated with the basal insulinemia (Figure 6D), in line with our findings in the CBS+/− mice, which had an insulin secretion defect (Spearman correlations *p*-values ≤ 0.05), while no significant correlation was seen with insulin resistance.

Next, a network was created with five genes involved in the homocysteine metabolism expressed in the pancreatic islets (*Cbs, Dyrk1a, Mat1a, Mthfr, Mtr, Ahcy*), their gene co-expression modules, the selected annotated KEGG pathways/GO categories of the modules, and the phenotypic traits represented as nodes (Figure 6E). We identified that the genes *Mtr*, *Dyrk1a*, and *Cbs* were co-expressed with genes implicated in the β-cell function such as the insulin/IFG1R signaling pathway and genes that encode for the complex I, II, III, IV, and V subunits of the oxidative phosphorylation. *Ahcy* is linked to genes participating in the cell signaling such as the phosphatidylinositol and the PI3K/Akt signaling pathways. Appendix A summarizes the Spearman correlations for the six genes involved in the metabolism of homocysteine and the phenotypic traits.

## 4. Discussion

HHcy has been demonstrated to promote insulin resistance in both humans and rodents [7], however, the consequences of an inborn defect in an enzyme in sulfur amino acid metabolism, CBS, on glucose homeostasis in mice have not been described.

Through the use of three mouse strains on HFD, transcriptomic analysis identified that the CBS gene expression in the islet correlated positively with insulin resistance and basal insulinemia. In the liver, the correlation with basal insulinemia was also positive. We thus postulated that HHcy due to CBS deficiency could be associated with decreased insulin secretion. We used a mouse model with CBS deficiency, CBS+/− mice, on a regular and methionine enriched diet to see the effect of moderate to intermediate HHcy. Interestingly, the defect in stimulated insulin secretion was evidenced in mice with moderate and intermediate HHcy, while mice with intermediate HHcy under HFD presented an improvement in insulin sensitivity (compared to HFD CBS+/+), which compensated for the decreased insulin secretion and permitted the glucose tolerance to be maintained similar to the CBS+/+ mice.

Previous studies have shown the detrimental role of Hcy on insulin secretion and pancreatic beta-cell function. The in vitro analysis showed an impairment in the insulin secretory function of the BRIN-BD11 beta-cells by Hcy in a concentration-dependent manner [27,28]. Hcy generates reactive oxygen species in a redox-cycling reaction with alloxan, a beta-cell toxin, which explains the decline in the viability of insulin-secreting cells, leading to reduced glucokinase phosphorylating ability, diminished insulin secretory responsiveness, and cell death [28].

Hydrogen sulfide (H_2_S) is an endogenous gastrotransmitter in mammalian cells. It can be generated from L-cysteine and/or Hcy and is catalyzed by two enzymes, CBS and cystathionine gamma-lyase (CGL) [29]. Endogenous H_2_S can be synthesized by multiple systems, with the highest rates occurring in the brain, cardiovascular system, liver, and kidney. Plasma H_2_S concentration has been found to be decreased in diabetic subjects compared with controls as well as in streptozotocin-treated diabetic rats compared with control Sprague-Dawley ones. The stimulatory effect of H_2_S on insulin secretion has been reported in vitro by Takahashi et al. [30], who showed that the inhibition of CBS reduced the cysteine hydropersulfide levels and decreased the glucose-induced insulin release in two different β-cell lines [30]. In the central nervous system, the CBS/H_2_S pathway in the paraventricular nucleus improved obesity and insulin sensitivity by regulating the neuroendocrine hormones via the brain–adipose interaction in the HFD rats [31]. The above studies showed that CBS/ H_2_S may play a role in the development of diabetes. In our study, we used mice deficient in CBS in all organs, and we previously showed a significantly lower CBS expression in the hypothalamus [9]. Although we did not assay the H_2_S, we can speculate that decreased insulin secretion could be related to lower CBS activity in the brain and autonomic regulation. In our study, islets isolated from the CBS+/− mice maintained their ability to respond to elevated glucose levels by showing an enhanced oscillatory response in intracellular free Ca^2+^. No differences in the apparent beta-to-beta cell connectivity were detected in comparison to the CBS+/+ islets. Thus, the in vivo defect in the stimulated insulin secretion is likely to be driven by the autonomic nervous system.

The central nervous system is a key player in the regulation of energy homeostasis [32] and islet hormone secretion [33,34]. The autonomic nervous system (ANS) innervating pancreatic islets comprises parasympathetic and sympathetic nerves that release acetylcholine and catecholamines, respectively, whereas the activation of the beta-cell M3-muscarinic receptors by acetylcholine promotes insulin secretion [35], and adrenergic receptor activation by epinephrine inhibits insulin secretion [36]. The ANS also controls insulin target tissues such as liver, muscle (mostly under sympathetic control), and adipose tissues, and unbalanced ANS plays a causal role in the setting of metabolic syndrome [37]. In the islets of CBS deficient mice, we found that the gene expression of the alpha 2 adrenergic receptor and M3 muscarinic receptor was decreased. Physiopathological features of HHcy are similar to some of the physiopathological features found in sympathetic overactivity in the cardiovascular system. HHcy provokes an activation of the sympathetic system, thus contributing to vascular and endothelial damage [38]. In rats, a moderate state of HHcy was accompanied by an increase in thee systolic and diastolic blood pressure and associated with an increase in the sympathetic modulation and a decrease in the parasympathetic modulation [39]. Our results are thus in agreement with previous results showing that HHcy might modulate the nervous system and lead to an increased sympathetic/parasympathetic ratio that inhibits insulin secretion.

Patients with impaired glucose tolerance (IGT), an intermediate stage between normal glucose tolerance and T2DM and characterized by insulin resistance, have higher Hcy levels compared to subjects with normal glucose tolerance [40]. Here, we found that mice with HHcy due to CBS deficiency had lower insulin secretion, which was compensated for after one month of a high-fat diet with an improved insulin sensitivity. Taken together, these results support previous studies that have reported increased levels of Hcy in patients with T2DM [41,42]. A significant negative correlation between the plasma Hcy levels and insulin levels was found in a population that was studied 10 years apart [43].

Accumulating evidence suggests that the brain–islet axis also regulates beta-cell proliferation. It has been demonstrated that beta-cell proliferation is stimulated by the parasympathetic and is inhibited by sympathetic signals ex vivo and in vivo [44]. In our model, we found similar pancreas histomorphology in the CBS+/− Met-HFD mice compared to the controls, showing that the potential nervous changes due to CBS deficiency did not impair islet neogenesis nor adult beta cell mass. However, we found that islet gene expression of Pax 6, pdx1, Rfx6, and Nkx6, and Cx36 was decreased in the CBS+/− mice. These transcription factors are important regulators of beta-cell differentiation. In immortalized beta-cell lines, Pax6 was shown to bind and activate the promoters of insulin [45]. Its deletion in adult murine islet cells causes hyperglycemia, with reduced expression of insulin [46], a process that was attributed to the loss of beta-cell function and the expansion of alpha-cells [47]. We also found an increased glucagon expression, a hallmark of diabetes, which led to worsening of the hyperglycemia in the islet of mice deficient in CBS. In the same line, the reduction in Pdx1 mRNA and protein levels was shown to be accompanied by deficiencies in the islet beta-cell function, beta-cell proliferation, and whole-body glucose homeostasis [48]. In addition, the transcription factors Neurogenin3, Rfx6, and Nkx6 also showed a lower expression in the islets of CBS+/− mice, indicating that beta-cell differentiation was incomplete, and these findings are in line with the insulin secretion defect [49,50]. The mRNA expression levels related to insulin biosynthesis (Ins2, Mafa, pdx1) were significantly decreased in the islets of the CBS deficient mice. Taken together, these results demonstrate that CBS plays an important role in the maintenance of beta-cell differentiation and function.

The network transcriptomic analysis of mice on HFD also revealed the implication of other genes implicated in Hcy metabolism in insulin resistance. SAHH and Dyrk1A gene expression in the islet correlated negatively with insulin resistance. We previously demonstrated a positive correlation between DYRK1A and SAHH activity in different organs of mice [51,52]. DYRK1A is expressed in the peripheral organs and has been shown to be related to diabetes phenotypes, while in wild-type mice, a HFD induces insulin resistance, which is not the case in mice overexpressing DYRK1A [16]. Consistent with their overall improved metabolic phenotype under HFD, these mice have increased beta-cell mass relative to wild-type mice. These results indicate that mBACTgDyrk1A mice overexpressing DYRK1A are resistant against HFD-induced diabetes [16]. Accordingly, Dyrk1A haploinsufficiency in mice produces severe glucose intolerance, hyperglycemia, hypoinsulinemia, and obesity, but without insulin resistance on a regular diet [53].

In conclusion, our results emphasize the important role of the CBS enzyme and the homocysteine pathway in controlling both the insulin secretion and insulin action. Altogether, our findings highlight the importance of monitoring homocysteine metabolism in prediabetic patients, and also suggest that people suffering with HHcy should carefully monitor their glucose homeostasis parameters. This “symmetrical” approach could lead to new personalized medical and nutritional advice, for example, patients with T2D should check their Hcy levels and eventually follow a specific diet poor in methionine to prevent any increase that could worsen their medical condition.

## Figures and Tables

**Figure 1 cells-11-01737-f001:**
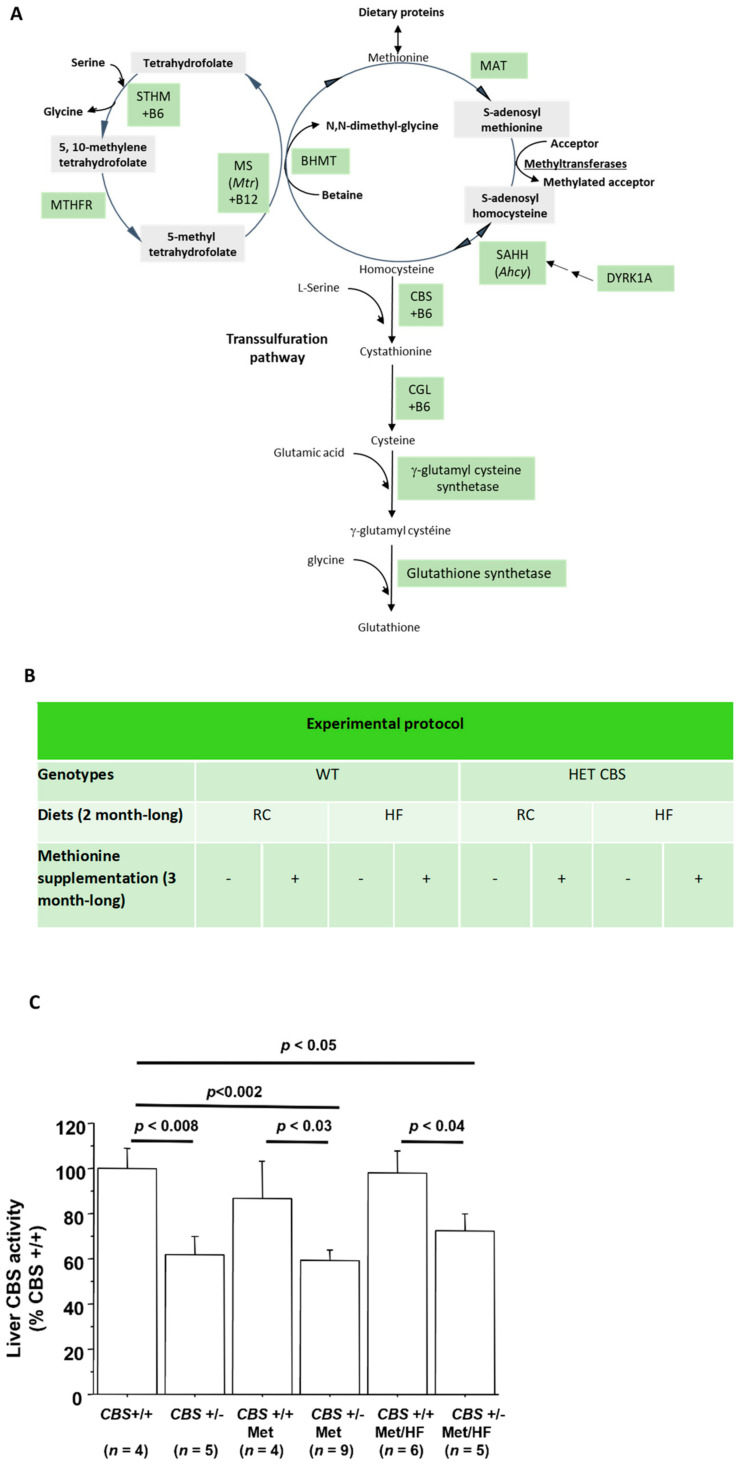
Targeting the homocysteine metabolic pathway. (**A**) The metabolic pathway of homocysteine. (Adapted from Le Stunff et al. [9]). (**B**) The mice experimental protocol. (**C**) The liver CBS activity.

**Figure 2 cells-11-01737-f002:**
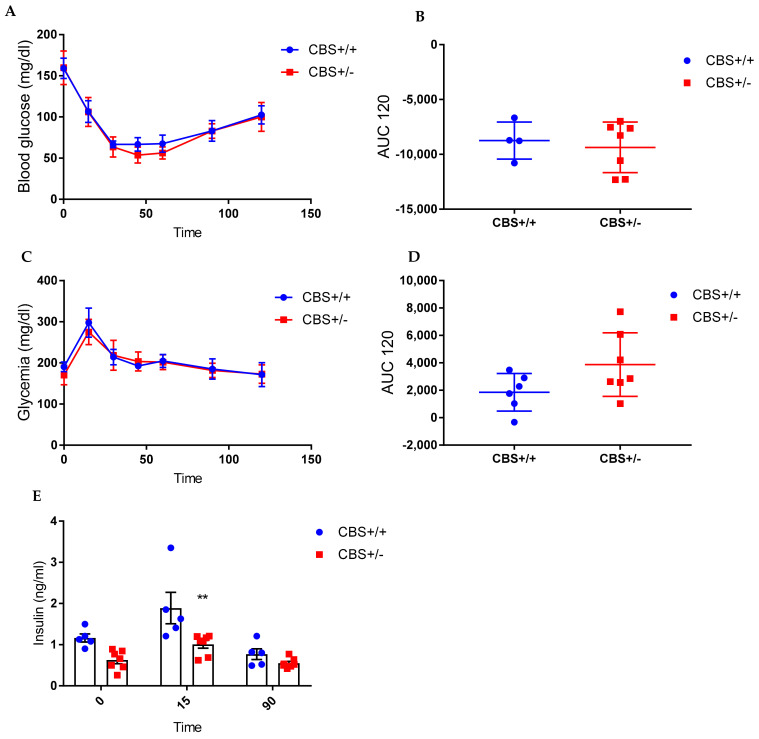
Energy homeostasis in the CBS+/− mice (genotype effect). (**A**) The insulin tolerance test (ITT) 0.5 UI/kg bw. (**B**) The area under the curve (AUC) of the ITT. (**C**). Oral glucose tolerance test 2 g/kg bw. (**D**) The AUC of OGTT. (**E**) Insulin secretion at baseline and in response to OGTT. Data are Means ± SEM. ** *p* < 0.01 vs. CBS+/+. Analysis by two-way ANOVA with post hoc Bonferroni test.

**Figure 3 cells-11-01737-f003:**
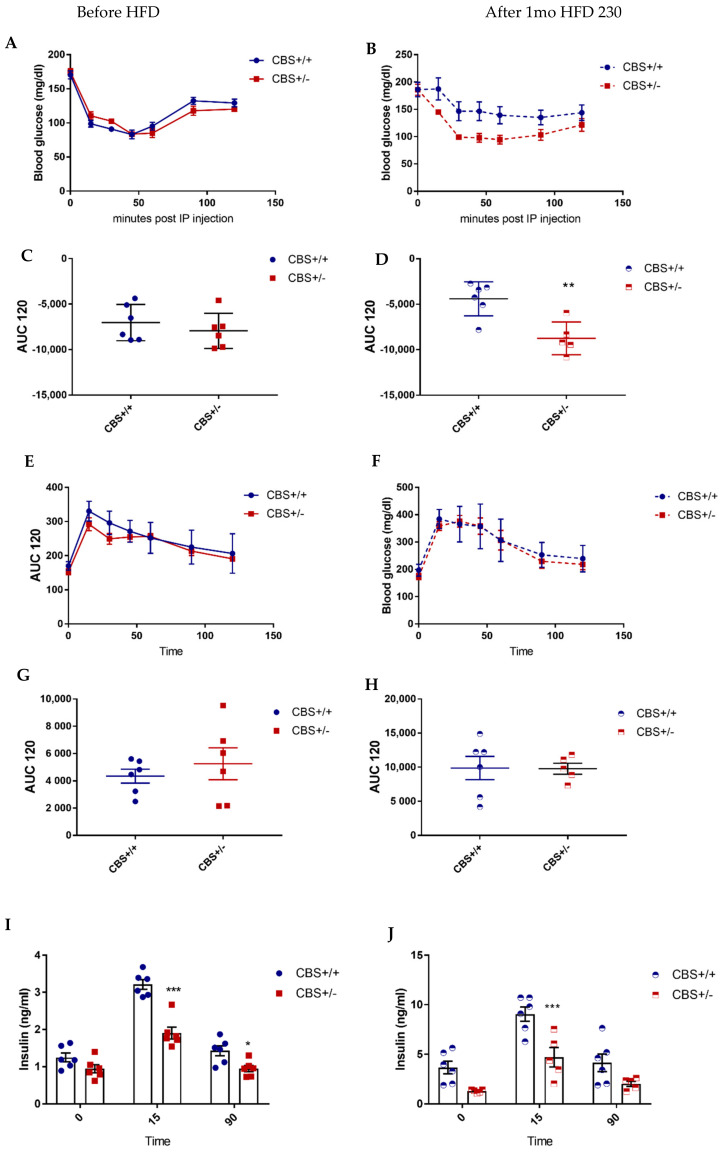
Energy homeostasis in CBS+/− mice supplemented with methionine and fed with HFD. Left panel: Study in mice supplemented with methionine and fed with a regular diet. (**A**) Insulin tolerance test (ITT) 0.5 UI/kg bw. (**C**) The area under curve (AUC) of ITT. (**E**) Oral glucose tolerance test 2 g/kg bw. (**G**) The AUC of OGTT. (**I**) Insulin secretion at the baseline and in response to OGTT. Right panel (**B**,**D**,**F**,**H**,**J**): same measurements in mice supplemented with methionine and fed with a high-fat diet, Met-HF mice. Data are Means ± SEM. ** *p* < 0.01, *** *p* < 0.005 vs. CBS+/+. Analysis by two-way ANOVA with post hoc Bonferroni test, and unpaired *t*-test for AUCs.

**Figure 4 cells-11-01737-f004:**
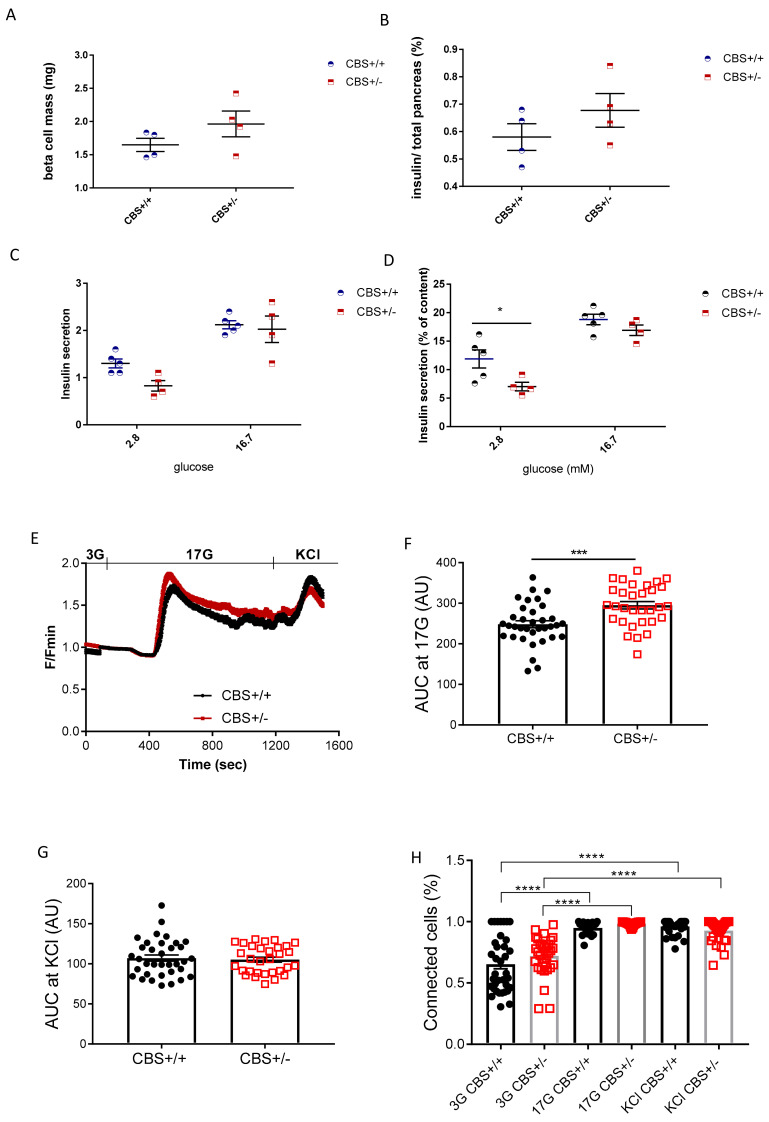
In vitro study in Met-HF mice. (**A**) Beta cell mass (mg). (**B**) Percentage of insulin reported to total pancreas. (**C**) In vitro insulin secretion of the isolated islets from the CBS+/+ and CBS+/− mice. (**D**) Normalization of insulin secretion by insulin content. (**D**) Intracellular Ca^2+^ responses from the CBS+/+ (*n* = 34) and CBS+/− (*n* = 31) islets isolated from the three mice per genotype in response to low (3 mm), high (17 mm) glucose, or 20 mm KCl. Both fluorescence amplitude.(* *p* < 0.05) (**E**) and (**F**) the area-under-the-curve of the glucose-evoked Ca^2+^ traces were increased in the CBS+/− islets (AUC; *** *p* < 0.001) under high glucose but not under the depolarizing stimulus with KCl (**G**). (**H**) The percentage of connected beta-cell pairs increased in both groups in response to 17 mm glucose or KCl (**** *p* < 0.0001). No differences were depicted between the mouse groups. Data are presented as Means ± SEM. Analysis by the unpaired *t*-test and two-way ANOVA with Tukey’s test.

**Figure 5 cells-11-01737-f005:**
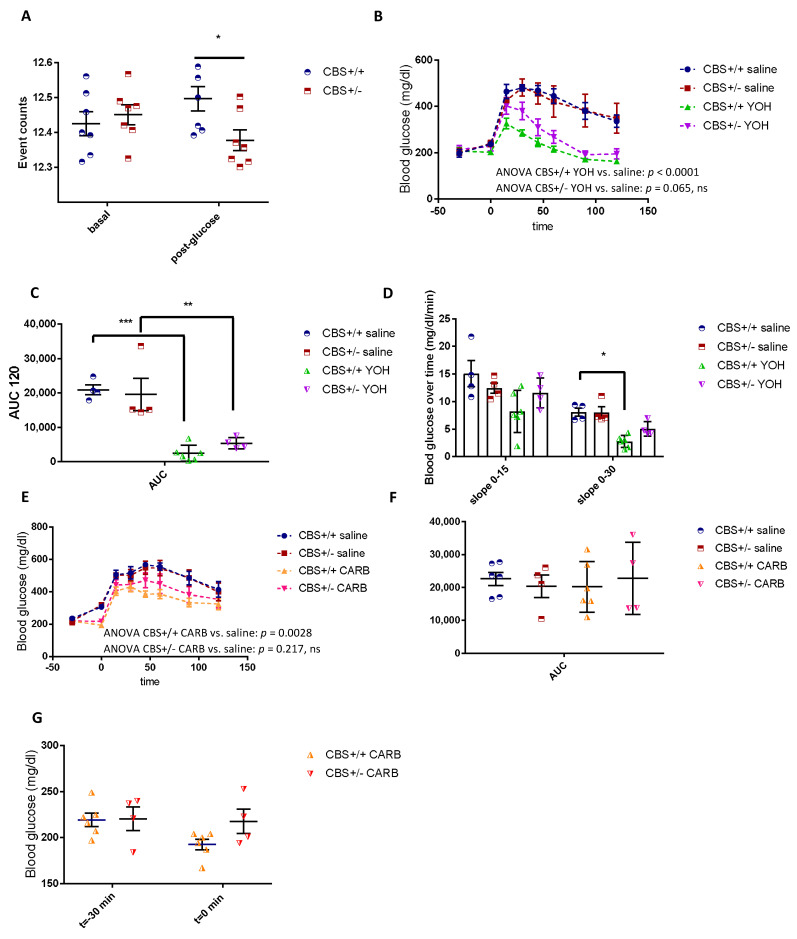
The nervous recording and autonomic nervous system modulation. (**A**) The PNS recording of the signals between 1 and 30 Hz, basal and post IP glucose administration. (**B**) The OGTT after the administration of yohimbine or vehicle. (**C**) The AUC of OGTT. (**D**) The blood glucose variation over time. (**E**) The OGTT after the administration of carbachol or the vehicle. (**F**) AUC of OGTT. G. Carbachol effect on basal blood glucose. Data are Means ± SEM. * *p* < 0.05, ** *p <* 0.01, *** *p* < 0.001 vs. (**G**) CBS, +/+ Analysis by two-way ANOVA with post hoc Bonferroni test, and unpaired *t*-test for AUCs.

**Figure 6 cells-11-01737-f006:**
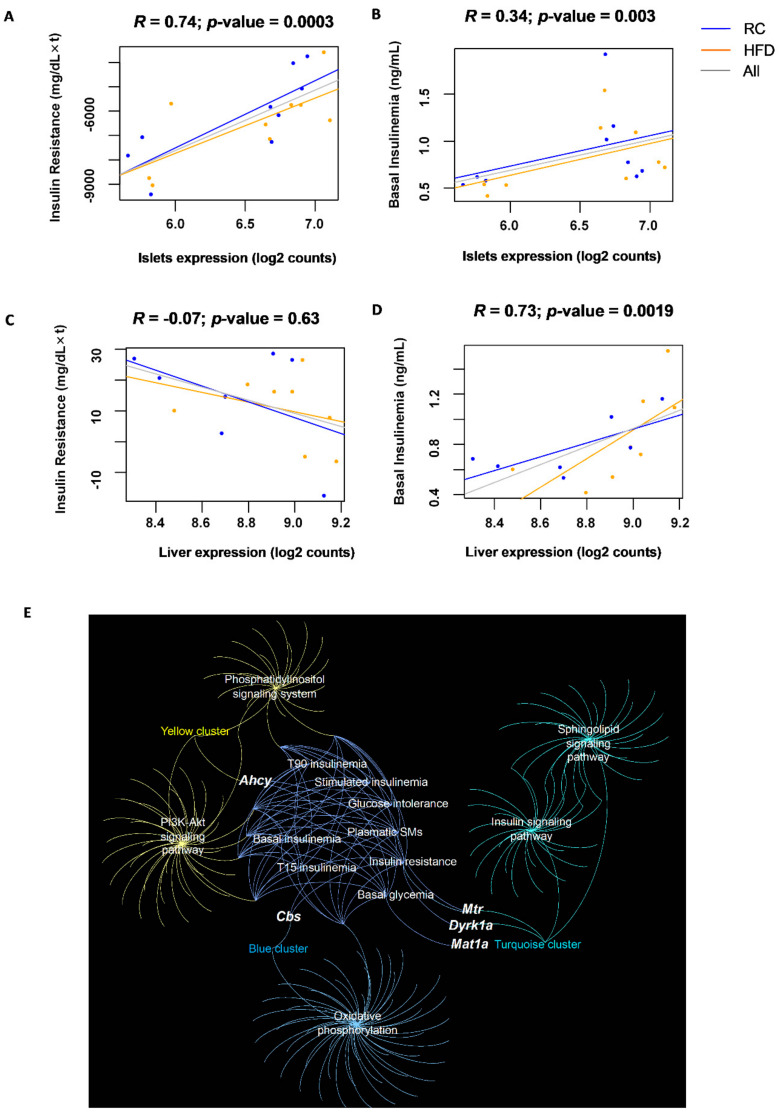
The pancreatic islet correlations and liver correlations in the Rhapsody cohort of 3 strains of mice fed with RC or HFD. (**A**) Correlation of the CBS pancreatic islet expression and insulin resistance assessed by the AUC of ITT. (**B**) Correlation of the CBS pancreatic islet expression and basal insulin secretion. (**C**,**D**) Same correlations in the liver. (**E**) Network representing genes involved in homocysteine metabolism in the islets, their gene co-expression modules, the annotated KEGG pathways/GO categories of the modules, and the phenotypic traits represented as nodes.

**Table 1 cells-11-01737-t001:** Hcy levels, body weight, fat and lean mass depending on the genotype and treatment. Data are Means ± SEM, *n* = 6 per group. *** *p* < 0.005 vs. respective CBS+/+.

Genotype	CBS+/+	CBS+/−	CBS+/+	CBS+/−	CBS+/+	CBS+/−
Age	2 mo	2 mo	7 mo	7 mo	8 mo	8 mo
Treatment and diet	none	none	methionine for 5 mo	methionine for 5 mo	Meth. for 6 mo, HFD for 1 mo	Meth. for 6 mo, HFD for 1 mo
Hcy (µmol/L)	3.3 ± 0.3	7.4 ± 1 ***	9.6 ± 2.7	30.4 ± 4.6 ***	6.4 ± 0.4	37.1 ± 3.2 ***
Body weight (g)	28.3 ± 1.74	27.53 ± 0.66	33.50 ± 0.44	31.53 ± 1.09	42.17 ± 1.1	37.98 ± 1.78
Fat mass (g)	4.51 ± 0.35	4.56 ± 0.42	7.06 ± 0.45	6.01 ± 0.73	16.36 ± 1.04	13.34 ± 1.34
Lean mass (g)	21.51 ± 1.42	20.63 ± 0.64	23.89 ± 0.34	22.93 ± 0.38	23.43 ± 0.37	22.47 ± 0.45

**Table 2 cells-11-01737-t002:** Gene expression in the pancreatic islets. * *p* < 0.05, ** *p* < 0.01, *** *p* < 0.001 vs. CBS+/+.

Pancreatic Islets (mRNA)	CBS+/+	CBS+/−
	(*n* = 4)	(*n* = 3)
AdRa2a (adrenergic receptor)	100 ± 16	43 ± 5 **
Cbs	100 ± 16	58 ± 14 *
Cx36 (connexin 36)	100 ± 12	30 ± 10 ***
Gck	100 ± 11	78 ± 19
Glucagon	100 ± 26	262 ± 17 ***
Glut2	100 ± 40	248 ± 87 *
Ins1	100 ± 7	139 ± 33
Ins2	100 ± 4	76 ± 10 **
Ldha	100 ± 20	53 ± 22
MafA	100 ± 13	30 ± 7 ***
Mct1	100 ± 10	61 ± 16
M3 (muscarinic Receptor)	100 ± 20	40 ± 15 *
Nkx6-1	100 ± 15	68 ± 6
Nkx6-2	100 ± 20	26 ± 3 **
Pax6	100 ± 12	65 ± 11 *
Pdx1	100 ± 7	51 ± 4 ***
Rfx6	100 ± 11	70 ± 7 *
Somatostatin	100 ± 15	69 ± 15

## Data Availability

Not applicable.

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
