# Peer review of "Homocysteine Metabolism Pathway Is Involved in the Control of Glucose Homeostasis: A Cystathionine Beta Synthase Deficiency Study in Mouse"

_cells, 2022, doi:10.3390/cells11111737_

Round 1

Reviewer 1 Report

In this manuscript, Cruciani-Guglielmacci et al. provide new information on the role of hyperhomocysteinemia (because of dietary manipulation or CBS inborn defects) in the regulation of insulin secretion and risk of progression to T2DM.
The study is metodologically sound, well written and suitable for publication. I would only suggest to enhance the resolution of figure 2. 

Regards

Author Response

We thank the reviewer for her/his comment. We have enhanced the resolution to figure 2, and we have also increased the size of the graphs for more legibility.

Reviewer 2 Report

The paper entitled “Homocysteine metabolism pathway is involved in the control of glucose homeostasis: a cystathionine beta synthase deficiency study in Mouse” includes potentially relevant data for the molecular diagnostics of metabolic disorders with underlying glucoce homeostasis imbalance as well as for the pharmacology of obesity and/or diabetes. Authors of this publication suggest that Islets isolated from CBS+/- mice maintain their ability to respond to elevated glucose levels, as well as that a lower parasympathetic tone could, at least in part, be responsible of the insulin secretion defect.

Remarks:

  1. Table no. 1 is illegible – it strongly needs to be modified.
  2. The resolution of the figures no. 2, no. 4E is inadequate – it should be improved.
  3. The rationale for the choice of the animal model should be provided
  4. The Conclusions section is too vague – it might be remodeled.
  5. Authors stated (Discussion section lines 428-429): The central nervous system is a key player in the regulation of energy homeostasis [31] and islet hormone secretion [32]. However, in the article no attempts can be to connect the Central Nervous System - its role in maintaining glucose homeostasis and insulin resistance of tissues - hepatic and adipose. If possible and if the Authors are in possession of relevant information - adding it - would significantly increase the scientific attractiveness for the potential reader.
  6. In the materials and methods section, the “2.7. Static batch incubation” subsection lacks information about the standardization of the procedure - it should be completed
  7. In the materials and methods section, the “2.7. Static batch incubation” subsection lacks information about the standardization of the procedure - it should be completed
  8. In the materials and methods section, the “2.11. Network construction” subsection lacks information about the standardization of the procedure - it should be completed
  9. In the discussion - in the conclusions chapter, there is no information about the practical / clinical / medical implications - the potential application of the obtained research results. Therefore, The Authors should clearly indicate the clinical implications of the presented results - if it is possible at the presented stage.

Author Response

We thank the reviewer for her/his comments. We have amended the manuscript accordingly, as detailed below:

  1. We have modified Table 1 for more legibility, in particular we added columns limits, and we removed the colour filling.

The resolution is high in our computer, in case of compatibility issues we propose to upload the original excel file, if needed.

  1. We have enhanced the resolution to figure 2 and 4E, and we have also increased the size of the graphs for more legibility (including Figure 3 and 4).

  1. Rodents are commonly used to study energy metabolism as mice and rats are susceptible to develop obesity and glucose intolerance under high-fat and high-sucrose diet (cf. Ref 11, Cruciani-Guglielmacci et al. 2017), thus mimicking the human aetiology of type 2 diabetes. In addition, as we stated in the introduction l. 45, a deficiency in CBS activity led to increased plasma homocysteine levels in both human and rodents. Thus, CBS heterozygous mice, which display lower CBS activity, appear to be an interesting animal model to question the link between homocysteine level and glucose homeostasis. The supplementation with 0.5% L-methionine in drinking water allows to induce intermediate HHcy in CBS +/− mice, and potentially increase the phenotype (as shown in Table 1, Hcy levels were 4-fold increased due to methionine supplementation: 7.4 µmol/l before, 30.4 µmol/l after). This latter model (CBS HET + Methionine) has been used in previous publications (Le Stunff et al., 2019; Baloula et al., 2018, respectively ref. 9 and 10).

We have modified the text l. 81-82 in the introduction.

  1. We agree that the conclusion section was short and vague, and we remodelled it substantially, in particular by taking into account your last point (9.) about clinical and practical implications of the results.

The changes are visible page 21 lines 512-520.

  1. Indeed, we agree that there is no direct link shown in this article between the lack of activation of the parasympathetic nervous system in response to glucose (shown in Figure 5A) and the liver production of glucose nor specific activation of adipose tissue. However, the parasympathetic nervous system (PNS) is well known to drive anabolism, in particular through its stimulatory effect on beta cell insulin release. On the contrary sympathetic nervous system (SNS) exerts a brake on insulin secretion. We have added a new reference, Rosario et al. 2016, which describes the “brain-to-Pancreatic islet neuronal map” (ref 34).

In addition, the autonomic nervous system (ANS) regulates the function of insulin target tissues, for example glycogenogenesis is stimulated in liver by the PNS and decreased by SNS activation which promotes glycogenolysis. We have partly rewritten this paragraph l. 458-460 to take in account the “extra pancreas” control, and inserted a new reference, Kreier et al. 2003 (now ref 37) which shows that unbalanced autonomic nervous system is linked to metabolic syndrome, and that ANS ratio between SNS and PNS could be differentially modulated in intra-abdominal compartment and in muscle.

In addition, we do have an information in the glycolytic muscle EDL (Extensor Digitorum Longus): as shown in Supplemental Figure 1, the glucose uptake is increased in CBS HET mice, suggesting an increased insulin sensitivity in this type of muscle. It would be interesting (but difficult in such a little animal as the mouse is) to record specific nervous activity towards pancreas, liver and muscle. We could nevertheless speculate that the increased glucose uptake in muscle is partly due to autonomic regulation.

  1. We have completed this subsection by giving additional information.
  2. (Same as 6)

  1. We have also completed the Network construction subsection and added a reference (Robinson et al. 2010, ref 20).

  1. We thank the reviewer for her/his suggestion to add information about practical/clinical/medical implication of our results. Our findings indeed suggest that people suffering with HHcy should carefully monitor their glucose homeostasis parameters, and conversely for T2D patients who should check their Hcy levels and, eventually, follow a specific diet poor in methionine to prevent any increase which could worsen their medical conditions.

We have added a paragraph about this point, to implement the conclusion section (also following #4.), line 512-520.

Round 2

Reviewer 2 Report

  The authors modified the manuscript, adapting the publication to the recommendations of the reviewers. The publication is acceptable without amendment.